# The Characteristics of Tumor Microenvironment Predict Survival and Response to Immunotherapy in Adrenocortical Carcinomas

**DOI:** 10.3390/cells12050755

**Published:** 2023-02-27

**Authors:** Guichuan Lai, Hui Liu, Jielian Deng, Kangjie Li, Cong Zhang, Xiaoni Zhong, Biao Xie

**Affiliations:** Department of Epidemiology and Health Statistics, School of Public Health, Chongqing Medical University, Yixue Road, Chongqing 400016, China

**Keywords:** adrenocortical carcinoma, tumor microenvironment, immunotherapy, subtype, bioinformatics, risk stratification, prognosis prediction

## Abstract

Increasing evidence confirms that tumor microenvironment (TME) can influence tumor progression and treatment, but TME is still understudied in adrenocortical carcinoma (ACC). In this study, we first scored TME using the xCell algorithm, then defined genes associated with TME, and then used consensus unsupervised clustering analysis to construct TME-related subtypes. Meanwhile, weighted gene co-expression network analysis was used to identify modules correlated with TME-related subtypes. Ultimately, the LASSO-Cox approach was used to establish a TME-related signature. The results showed that TME-related scores in ACC may not correlate with clinical features but do promote a better overall survival. Patients were classified into two TME-related subtypes. Subtype 2 had more immune signaling features, higher expression of immune checkpoints and MHC molecules, no CTNNB1 mutations, higher infiltration of macrophages and endothelial cells, lower tumor immune dysfunction and exclusion scores, and higher immunophenoscore, suggesting that subtype 2 may be more sensitive to immunotherapy. 231 modular genes highly relevant to TME-related subtypes were identified, and a 7-gene TME-related signature that independently predicted patient prognosis was established. Our study revealed an integrated role of TME in ACC and helped to identify those patients who really responded to immunotherapy, while providing new strategies on risk management and prognosis prediction.

## 1. Introduction

Adrenocortical carcinoma (ACC) originates from the adrenal cortex and is a highly invasive tumor of the endocrine system with an annual incidence of approximately 0.5–2 cases per million people. ACC can occur at any age, has a peak incidence in the 40 s and 50 s, and a higher rate of diagnosis in women than in men [1,2]. The primary treatments available for ACC are surgical resection, mitotane therapy, chemotherapy, radiotherapy, and immunotherapy [3,4,5,6,7]. Most patients have difficulty achieving a complete cure due to the adverse effects of ACC, such as high post-operative recurrence rates, low response to drug treatments, and toxic side effects of the drugs [8,9,10]. Although the incidence of the disease is low, patient survival is poor, with an overall survival (OS) rate of less than 15% at 5 years [11]. Age, tumor stage, tumor grade, and cortisol secretion have been reported to be the main factors affecting patient prognosis [12,13,14,15]. Although these disease markers can help us to predict patient prognosis to some extent, there is still a need for more and more accurate biomarkers used in predicting patient survival and treatment outcome. With the rapid development of multi-omics in tumors, gene markers, mutations in genes, and involved pathways were closely related to the biological functions of ACC and had the promise to become its potential therapeutic targets [16]. Kamilaris et al. found that overexpression of IGF2, mutations in TP53, ZNRF3, CTNNB1, and 11p15, and abnormal alterations in WNT/β-catenin and p53 signaling pathways contributed to the development of ACC [17]. Meanwhile, the rational development of targeted predictive models for genetic markers, the establishment of risk stratification for treatment, the development of therapeutic agents targeting pathways, and the conduct of immunotherapy trials are the main tasks at hand [18].

The tumor microenvironment (TME) is a complex multicellular environment in which the tumor is located, primarily composed of immune cells, stromal cells, an extracellular matrix, secreted small molecules, and blood and lymphatic vascular networks, and is highly heterogeneous and dynamic [19]. The development of TME is dedicated to tumor progression; its main components play an important role [20]. These include tumor-associated macrophages, myeloid-derived suppressor cells, regulatory T cells, extracellular matrix, and tumor vasculature [21,22,23,24,25,26]. Immune cells (CD8^+^ T cells and mast cells) and vascular endothelial cells can promote or inhibit ACC progression through multiple pathways [27,28,29,30]. For example, high infiltration of CD8^+^ T cells were observed in younger and stage I patients, and were associated with a better prognosis [27]. A high abundance of tumor mast cells promoted both better OS and progression-free survival, enhancing the accumulation of CD8^+^ T cells and CD4^+^ T cells [28]. Endothelial cell contraction, migration, and proliferation are key factors for the occurrence of angiogenesis, that promoted tumor development to malignancy [29,30]. In fact, TME does not only affect tumor growth, but is equally important for immunotherapy. Studies have demonstrated that the heterogeneity of the TME affected the response of patients to immunotherapy, and some studies have found that TME-related subtypes helped to select patients who were responsive to immunotherapy [31,32,33,34,35,36]. However, there are still no studies on the relationship between TME-related subtypes and response to immunotherapy in ACC.

In summary, we used the xCell algorithm to quantify the microenvironment of ACC, starting with the TME-related score, and fully considered the role of TME-related subtypes in the immunotherapy of ACC. Finally, we established a novel TME-related signature that could achieve the risk classification and prognosis prediction of ACC, providing the scientific basis on clinical treatment for ACC in the future.

## 2. Materials and Methods

### 2.1. Data Collection and Pre-Processing

We downloaded transcriptomic data and clinical information of ACC from The Cancer Genome Atlas (TCGA) (https://xenabrowser.net/, accessed on 1 November 2022) website. This tool was used to determine gene expression profiles through the Illumina platform. In this study, FPKM values were converted to transcript per million (TPM), and gene expression was estimated primarilyin log_2_(TPM + 1). We obtained the“Homo_sapiens.GRCh38.108.chr.gtf.gz” file from the Ensembl (http://Asia.ensembl.org/, accessed on 2 November 2022) database for gene annotation. For the clinical data in the TCGA-ACC dataset, samples with complete gene expression and OS information were enrolled in this study. Based on this condition, we obtained a total of 79 subjects from the TCGA-ACC dataset. The primarily clinical indicators included age, gender, T stage, N stage, M stage, and tumor stage. In addition, GSE10927 and GSE33371 were downloaded from the GEO (http://www.ncbi.nlm.nih.gov/geo/, accessed on 3 November 2022) database and the original expression matrix in “CEL” format was background corrected and normalized using the Robust Multichip Average algorithm [37]. Clinical information was downloaded using the “GEOquery” package [38]; a total of 47 subjects were included from the GSE10927 and GSE33371 datasets according to the filtering criteria, including age, gender, Weiss grade, and tumor stage. In this study, we used the TCGA-ACC dataset as the training set and the GSE10927 and GSE33371 datasets as the external validation set to validate the predictive ability and risk stratification of TME-related signature. The detailed flow chart was shown in Figure 1.

### 2.2. Quantifying the Tumor Microenvironment

The xCell algorithm was based on scoring and specifically transforming the 489 reliable gene sets of 64 cells using the principles of the ssGSEA algorithm [39]. We used the xCell algorithm to obtain 64 cells from 79 samples, as well as markers that could synthesize the corresponding characteristics of these cells: immune score, stromal score, and microenvironment score (estimate score). The immune score and stromal score reflected the overall profile of immune cells and stromal cells, respectively, while the microenvironment score, that assumed a negative correlation with tumor purity, was numerically equal to the sum of immune score and stromal score and could be used as a measure of TME.

### 2.3. Identification of TME-Related Genes

First, we used the “surv_cutpoint” function to determine the optimal cutoff values for the immune score, stromal score, and microenvironment score. Next, differential expressed analysis was performed between high- and low-score groups using the “edgeR” package (a differential expression analysis method based on count data [40]). Finally, we identified TME-related genes by intersecting up-regulated and down-regulated genes in the immune, stromal and, microenvironment groups.

### 2.4. Construction of TME-Related Subtypes

Based on the normalized expression of TME-related genes, ACC patients were classified into different TME-related subtypes through consensus unsupervised clustering analysis using the “ConsensusClusterPlus” package [41]. The seed number (123456) and the maximum number of clusters (k = 7) were established first, and the similarity was measured by Pearson method using hierarchical clustering. The clustering was completed by repeated sampling 1000 times with a resampling ratio of 80%. In this study, the optimal number of clusters was determined primarily based on the Proportion of Ambiguous Clustering (PAC) value, and the optimal k was determined when the PAC value was minimal [42].

### 2.5. GSVA Analysis of the KEGG and Hallmark Pathways among Different Subtypes

The Gene Set Variation Analysis (GSVA) algorithm is a commonly used method that can evaluate the enrichment score (ES) of a sample in a specific gene set [43]. In this study, 186 KEGG and 50 Hallmark pathways were downloaded from the Molecular Signatures Database (https://www.gsea-msigdb.org/, accessed on 7 November 2022), and the ES of these pathways in ACC were calculated using the GSVA algorithm. The “limma” package was used to screen the differential pathways among different subtypes, and the screening criterion was that the corrected *p*-value was less than 0.05 [44].

### 2.6. TME-Related Characteristics among Different TME-Related Subtypes

Based on the EPIC method, we calculated the abundance of B cells, cancer associated fibroblast (CAF) cells, CD4^+^ T cells, CD8^+^ T cells, endothelial cells, macrophage cells, and natural killer (NK) cells [45]. To ensure the stability of the results, we performed the same analysis using the xCell algorithm.

### 2.7. Somatic Mutation Analysis

Mutation data from ACC were downloaded using the “TCGAbiolinks” and “maftools” packages [46,47]. In this study, the exome length was defined as 40 mb. We used an estimate of the total number of somatic mutations divided by exome length to measure the TMB [48].

### 2.8. Immunotherapy Response among Different TME-Related Subtypes

The Tumor Immune Dysfunction and Exclusion (TIDE) and the immunophenoscore (IPS) algorithms were used to assess the ability of ACC patients to respond to immune checkpoint inhibitors (ICIs) [49,50]. Patients with high TIDE scores were generally considered to be more prone to immune escape and less amenable to ICIs, whereas tumor patients with low TIDE scores were more responsive and amenable to ICIs. The TIDE involved two main components: T cell dysfunction and exclusion. The IPS algorithm was composed of major histocompatibility complex (MHC) molecules, effector cells (EC), immune checkpoints (CP), and immunosuppressive cells (SC).

### 2.9. WGCNA Analysis

The median absolute deviation (MAD) value was used as a criterion to filter genes, and the genes with the top 5000 MAD values were subjected to Weighted Gene Co-Expression Network Analysis (WGCNA) [51]; the “goodSamplesGenes” function was used to check whether there were low-quality samples and genes. In this study, the optimal power could be estimated by the “pickSoftThreshold” function. The number of modules and categories were determined using the dynamic tree cut approach. The first component in each feature module was extracted using principal component analysis. The first component of each module could fully reflect the expression level of all genes in that module, and the correlation analysis of the first component of the characteristic gene module with the TME-related subtype was performed to identify the module with the highest correlation with the TME-related subtypes as the key module. These genes in the key module were regarded as TME-related subtype genes.

### 2.10. GO Enrichment and KEGG Signaling Pathway

We performed Gene Ontology (GO) enrichment and Kyoto Encyclopedia of Genes and Genomes (KEGG) pathway analyses on TME-related subtype genes. The GO enrichment included three aspects: molecular function (MF), biological process (BP), and cellular components (CC). We used the “clusterprofiler” package to implement the above process, and the screening criteria was *p*-value less than 0.05 after Benjamini–Hochberg correction [52].

### 2.11. LASSO-Cox Analysis to Construct and Validate a TME-Related Signature

Combining TME-related genes and TME-related subtype genes, we defined their intersected genes as candidate genes, and they were used for the construction of TME-related risk score. First, the candidate TME-related genes were subjected to univariate Cox analysis using the “survival” package, and genes with *p* < 0.05 were tagged as prognostic genes for ACC. These prognostic TME-related genes were then included in the least absolute shrinkage and selection operator (LASSO) regression analysis for further gene screening and risk score construction using the “glmnet” package; the lamda values were calculated using 10-fold cross-validation. Here, the lamda value with the smallest partial likelihood residuals was selected, and the genes with non-zero coefficients were retained as candidates for model construction. The risk score of ACC patients was calculated using the expression of linear regression. After obtaining the TME-related risk scores of patients in TCGA-ACC, GSE33371, and GSE10927 datasets, ACC patients were classified into high- and low-risk groups according to the median value of the risk score. The “timeROC” package was used to predict the area under the receiver operating characteristic curve (ROC) of ACC patients at years 1, 3, and 5. To test whether the risk score was independent of other clinical indicators used to predict OS, the clinical characteristics of patients were included in the TCGA-ACC, GSE33371, and GSE10927 datasets for univariate and multivariate Cox analyses, respectively, and variables with prognostic significance after univariate Cox regression were included in the multivariate Cox regression.

## 3. Results

### 3.1. TME and OS in ACC

Using the “surv_cutpoint” function, the cutoff values of immune score, stromal score, and microenvironment score were calculated as 0.028, 0.015, and 0.049, respectively. Based on the cutoff value, high-immune group (*N* = 32), low-immune group (*N* = 47), high-stromal group (*N* = 59), low-stromal group (*N* = 20), high-microenvironment group (*N* = 47), and low-microenvironment group (*N* = 32) were determined. High-stromal, high-immune, and high-microenvironment groups were associated with a better OS (Figure 2A–C).

### 3.2. TME and Clinical Features of ACC

We found that age was not statistically correlated with TME-related scores (Appendix A). Similarly, gender, T stage, M stage, and N stage were not statistically associated with TME-related scores according to the Wilcoxon test (Appendix A–L). Patients’ immune and stromal scores were also not statistically different in the tumor stage group (Appendix A), but microenvironment score showed a statistical significance (Appendix A).

### 3.3. Identification of TME-Related Genes

2977 differentially expressed genes (DEGs), including 2026 up-regulated genes and 951 down-regulated genes, were obtained between the high- and low-immune groups (Appendix A). In the stromal group, 2337 DEGs, including 1323 up-regulated and 1014 down-regulated genes, were screened (Appendix A). For the microenvironment group, 2965 DEGs were identified, including 1998 up-regulated and 967 down-regulated genes (Appendix A). The Venn diagram showed 730 up-regulated genes and 259 down-regulated genes at the intersection of the immune, stromal, and microenvironment groups (Figure 3). Thus, 989 TME-related genes were defined in this study.

### 3.4. Classification of TME-Related Subtypes

Based on the principle of PAC, we compared the magnitudes of cumulative distribution function (CDF) (0.1), CDF (0.9), and PAC for different k values. In Appendix A, the curve at k = 2 had a flatter change when the consistency index changed from 0.1 to 0.9. Meanwhile, the minimum PAC value appeared when k = 2, indicating that the optimal number of clusters is 2. Similarly, as k increased, the clustering effect became more unstable, with more ambiguous parts appearing, resulting in the inability to clearly delineate the different subtypes (Figure 4). Therefore, we classified the 79 subjects into two TME-related subtypes: subtype 1 (*N* = 47) and subtype 2 (*N* = 32). KM survival analysis showed that subtype 2 had a higher OS than subtype 1 (Figure 5).

### 3.5. Identification of Key Signaling Pathways between two TME-Related Subtypes

Based on the ES of 186 KEGG and 50 Hallmark pathways, a total of 102 differential KEGG pathways (Appendix A) and 26 differential Hallmark pathways between subtype 1 and subtype 2 were identified (Appendix A). Among the top 20 KEGG pathways, subtype 2 exhibited higher ES compared to subtype 1. These pathways mainly included the chemokine signaling pathway, NOD-like receptor signaling pathway, Toll-like receptor signaling pathway, Graft-versus-host disease, and Allograft rejection (Figure 6A). Among the 26 differential Hallmark pathways, subtype 1 was significantly enriched in nine signaling pathways, including MYC targets v1 and v2, G2M checkpoint, cholesterol homeostasis, and E2F targets, while subtype 2 was involved in 17 signaling pathways, primarily including Complement, Allograft rejection, Interferon gamma response, IL-2-STAT5 Signaling, and IL-6/JAK/STAT3 signalling (Figure 6B).

### 3.6. TME-Related Subtypes and Tumor Infiltration Cells

Using the EPIC algorithm, we evaluated the infiltrating levels of seven important cells. Our evaluation found the highest ratio to be CD4^+^ T cells and the lowest proportion to be NK cells (Figure 7A). Subtype 2 had higher levels of B cells, endothelial cells, and macrophage when compared to subtype 1 (Figure 7B). We also calculated the ratio of these seven cells using the xCell algorithm. The results showed that subtype 2 had a higher infiltrating levels of fibroblasts, CD8^+^ T cells, endothelial cells, and macrophages when compared to subtype 1 (Figure 7C). Subtype 2 had higher TME-related scores than subtype 1 (Figure 7D–F).

### 3.7. TME-Related Subtypes and Somatic Mutation

The mutational waterfall plot showed that the top 20 high mutated genes were TP53 (16%), CTNNB1 (15%), MUC16 (15%), TTN (11%), HMCN1 (9%), PKHD1 (9%), CNTNAP5 (8%), MEN1 (8%), MUC4 (8%), PRKAR1A (8%), ANK2 (6%), ASXL3 (6%), DST (6%), FAT4 (6%), NF1 (6%), STAB1 (6%), SVEP1 (6%), TMEM247 (6%), HIVEP1 (5%), and KMT2B (5%) (Appendix A). To investigate whether the mutation of these genes affected the OS, we compared the OS of patients with mutated genes to those without mutated genes. The results showed that only mutation in DST (*p* = 0.002), HIVEP (*p* = 0.028), ASXL3 (*p* = 0.001), PKHD1 (*p* = 0.012), TP53 (*p* = 0.003), and CTNNB1 (*p* = 0.049) was associated with OS, resulting in a worse prognosis (Appendix A). As for the mutational waterfall plot of subtype 1 and 2, we found a higher rate of CTNNB1 mutations in patients with subtype compared to those with subtype 2, using the corrected chi-square test (Figure 8A,B). By calculating the TMB of ACC patients, the median TMB was found to be 0.58/MB for overall ACC patients, 0.73/MB for patients with subtype 1, and 0.40/MB for patients with subtype 2 (Figure 9A–C). By comparing the TMB between subtype 1 and 2, it was shown that patients with subtype 1 had a higher TMB (Figure 9D). We classified TMB into high- and low-TMB groups according to median value of TMB. Patients in the high TMB group had a worse OS than those in the low TMB group (Figure 9E).

### 3.8. TME-Related Subtypes and Immunotherapy

In this study, we compared the expressions of six important MHC molecules and eight immune checkpoints (ICs) between subtypes 1 and 2. The results showed that they were all highly expressed in patients with subtype 2 (Figure 10 and Figure 11). Therefore, to understand whether TME-related subtypes affected the response to ICIs, the normalized TIDE scores and IPS of ACC patients were obtained using the TIDE and IPS algorithms. It was found that the normalized TIDE scores of subtype 2 were lower than those of subtype 1 and the IPS of subtype 2 were higher than those of subtype 1 (Figure 12), indicating that patients with subtype 2 may be more sensitive to ICIs than patients with subtype 1.

### 3.9. Identification of a Module Highly Correlated with TME-Related Subtypes

First, no abnormal samples and genes were found using the “goodSamplesGenes” function. The hierarchical clustering tree was drawn for 79 samples and no obvious outlier samples were found (Appendix A). As such, 5000 high MAD genes and 79 ACC subjects were retained for subsequent WGCNA analysis. The best soft threshold (β = 13) was obtained using the “pickSoftThreshold” function (Figure 13). Based on the determined optimal power, the 1-TOM matrix was obtained, hierarchical clustering was performed, and different modules were classified using the dynamic tree cut method, from which a total of 15 modules were obtained. The red module was found to be the most correlated with the TME-related subtypes (Figure 14), so 271 genes in this module were extracted for subsequent analysis (Appendix A). We defined these genes as TME-related subtype genes.

### 3.10. Biologically Functional Analyses of TME-Related Subtype Genes

To understand the biological functions of TME-related subtype genes, the most significant functions and signaling pathways were visualized by bubble plots (Figure 15 and Figure 16). In the BP, the five most significant functions were response to interferon-gamma, cellular response to interferon-gamma, leukocyte cell-cell adhesion, leukocyte chemotaxis, and regulation of mononuclear cell proliferation. In the CC, the five most prominent functions included MHC protein complex, MHC class II protein complex, integral component of lumenal side of endoplasmic reticulum membrane, lumenal side of endoplasmic reticulum membrane, and endocytic vesicle membrane. In terms of MF, the five most representative functions consisted of immune receptor activity, MHC class II receptor activity, peptide antigen binding, amide binding, and peptide binding. KEGG analysis showed that a total of 52 differential pathways were identified, of which the 10 most significant pathways were Staphylococcus aureus infection, Rheumatoid arthritis, Phagosome, Leishmaniasis, Viral myocarditis, Allograft rejection, Complement and coagulation cascades, Graft-versus-host disease, Type I diabetes mellitus, and Autoimmune thyroid disease.

### 3.11. Construction and External Validation of a TME-Related Signature

A total of 103 candidate TME-related genes were obtained by combining TME-related genes with TME-related subtype genes (Appendix A). Univariate Cox analysis of these genes revealed that 11 genes were associated with OS. Next, these prognostic genes were included in the LASSO regression analysis for the construction of a TME-related signature. The results showed that when the partial likelihood residuals were minimal, seven genes with non-zero coefficients were obtained and used to calculate a risk score (Figure 17A,B). In this study, the TME-related risk score was calculated as follows: 0.57572687 × Expression(RGS1) − 0.71891011 × Expression(CX3CR1) − 0.13089450 × Expression(CYTL1) − 0.15899458 × Expression(GBP2) − 0.03429512 × Expression(SLC40A1) − 0.04349504 × Expression(SLC9A9) − 0.02742250 × Expression(TFEB). In the TCGA-ACC dataset, the OS of high-risk patients was significantly lower than that of low-risk patients, with time-ROC curves showing AUC values of 0.91, 0.88, and 0.86 at years 1, 3, and 5, respectively (Figure 18). In the GSE10927 dataset, high-risk patients had a worse OS and the AUC values of years 1, 3, and 5 were 0.87, 0.77, and 0.77, respectively (Figure 19A,B). In the GSE33371 dataset, high-risk patients had an OS disadvantage and the AUC of 1, 3, and 5 years was 0.86, 0.75, and 0.75, respectively (Figure 19C,D). These results suggested that this TME-related signature had a better risk stratification ability and higher predictive power.

### 3.12. TME-Related Signature Was an Independent Prognostic Indicator

In the TCGA-ACC dataset, univariate Cox analysis showed that risk score, M stage, T stage, and tumor stage were significantly associated with OS (Table 1). Then, after including these significant variables from univariate Cox analysis into the multivariate Cox analysis, only risk score was found to be remarkably associated with OS, while the other variables were not statistically significant factors (Table 1). In both external datasets, we found that only risk score was a significant variable affecting OS (Table 2). Thus, these findings supported that this TME-related signature could be used as an independent prognostic indicator for ACC patients.

## 4. Discussion

Tumor development is closely related to the interaction between tumor cells and TME, and immune and stromal cells in the TME can promote or inhibit tumor growth. Therefore, it is important to understand the role of TME to inhibit tumorigenesis and metastasis.

In this study, we used the xCell algorithm to digitally quantify the TME, helping us understand the comprehensive impact of TME on tumors. We scored the immune and stromal microenvironment and identified the genes associated with them. In our study, all three TME-related scores were associated with the OS of ACC patients, indicating that TME-related characteristics were an influential factor in the prognosis of ACC patients. At the same time, none of the TME-related scores were statistically associated with most of the clinical features, suggesting that the clinical features of ACC patients did not affect the overall level of TME, further emphasizing the need to reveal TME at the molecular level. Using differential expressed analysis, we defined a total of 989 genes associated with TME.

In order to investigate whether TME-related subtypes had predictive potential in terms of immunotherapy. We first classified ACC into two TME-related subtypes based on TME-related genes using consensus unsupervised clustering analysis and defined them as subtype 1 and 2, respectively. KM survival analysis revealed that subtype 2 had a better OS than subtype 1, indicating that the potential ability of TME-related subtypes in the risk stratification was noticed.

In terms of signaling pathways, subtype 2 was closely associated with most immune-related pathways, including chemokine signaling pathway, NOD-like receptor signaling pathway, TOLL-like receptor signaling activity, Allograft rejection, Complement, interferon-gamma, IL2/STAT5, and IL-6/JAK/STAT3 signaling pathways. Chemokine expression was associated with T-cell inflammation in tumors [53]. CXCR4 and CXCR7 have been shown to be most abundant in ACC, yet they were not associated with prognosis in patients with ACC [54]. There were some NOD-like receptors involved in the formation of inflammatory vesicles, such asNLRP3, which was the most typical NOD-like receptor for inflammatory vesicle formation [55,56]. TLRs could mediate adaptive immunity, affecting T and B cell responses [57]. A pan-cancer analysis identified TLR4 and TLR5 as prognostic genes in ACC [58]. Also, immune-mediated upregulation of TLR4 signaling may become a novel strategy for immunotherapy of ACC [59]. The effect of alloantigens on innate immunity was associated with the allograft rejection pathway, while pro-inflammatory cytokines activated by innate immune-stimulated T cell expansion also occurred in this process [60,61]. In addition, the pathway was associated with the development of low-risk ACC patients [62]. Complement influenced the type and extent of the immune response by cooperating with other cellular defense pathways [63]. Interferon-gamma-induced signaling occurred in many types of immune cells, such as type I helper T cells, cytotoxic T cells (CTLs), and NK cells, while interferon-γ-dependent immunotherapy acted primarily through the anti-tumor mechanism [64]. Lower tumor purity may lead to overexpression of IL2/STAT5 signaling, while tumor purity has been shown to have a negative correlation with immune and stromal scores [65,66], suggesting that subtype 2 has more immune and stromal features relative to subtype 1;this is consistent with our results. The IL6-JAK-STAT3 signaling pathway could induce PD-1 and PD-L1 expression and promote anti-tumor immune effects [67]. Therefore, in our study, subtype 2 expressed more PDCD1 and CD274 than subtype 1, which may be related to the involvement of IL-6/JAK/STAT3 signaling pathway. In contrast, for subtype 1, MYC target v1 and v2, G2/M checkpoint, cholesterol homeostasis, and E2F target were the primary associated signals. A high ES of MYC target v1 andv2 pathways was found to be associated with cell proliferation in ER-positive/HER2-negative breast tumors, leading to a worse outcome [68]. G2/M checkpoint was a common signal in the cell cycle and was an important source for tumor survival [69]. Also, dysregulation of G2/M transition may affect the pathogenesis of ACC [70]. Cholesterol balance was essential for proper cell and system function [71]. It was found that mitotane affected the expression of genes involved in the homeostatic process of cholesterol, that in turn affected the prognosis of patients [72]. Enhanced E2F activity could promote the resistance to chemotherapy; some genes in this family may be involved in the progression of ACC, and E2Fs have been shown to be promising therapeutic targets for ACC [73,74]. Thus, the results suggested that the development of subtype 1 may be related to cell cycle activity, whereas the activity of subtype 2 may be more related to immune-related signals.

By comparing the tumor infiltrating cells of different TME-related subtypes, we found that endothelial cells and macrophages were significantly expressed in patients with subtype 2. Endothelial cells were able to produce more IFN-β, which was shown to inhibit ACC cell growth [75,76]; this may explain why there were more endothelial cells in patients with subtype 2. Macrophages played an important role in the immune microenvironment, with polarized pro-inflammatory M1 macrophages inhibiting tumor growth and anti-inflammatory M2 macrophages exerting a pro-tumor role [77]. M2 macrophages were the predominant macrophages in ACC, and lower infiltrating levels of M1 macrophages were found in cortisol-secreting ACC [78]. Recently, Guan et al. found that ACC1 had a higher level of macrophage infiltration compared to ACC2 and ACC3, and the OS of ACC1 was the highest among the three subtypes [79]. This finding was similar to the results of the present study. Mutations in tumor suppressor genes have emerged as one of the major causes of ACC pathogenesis [80]. TP53 and CTNNB1 had a highly mutated rate in overall ACC, while mutations in these two genes were associated with a poorer OS in patients. Mutations in TP53 were most prevalent in children with ACC and decreased over time, but the mutation carried a risk of familial inherited mutations [81,82]. CTNNB1 mutation activated the β-catenin pathway, and the presence of β-catenin nuclear staining caused by activation of the Wnt/β-catenin signaling pathway led to reduced OS and disease-free survival in ACC [83,84], supporting that CTNNB1 mutation was associated with a poor prognosis in this study. Most importantly, CTNNB1 mutation was not found in subtype 2. CTNNB1 mutation was found to be enriched in tumors with non-T-cell inflammation, and the activated Wnt/β-catenin signaling pathway was associated with immunosuppression [85]. Besides, a recent study on the immunophenotype of ACC showed that the low-immune group exhibited higher CTNNB1 mutations when compared to the high-immune group [86]. These findings suggested that subtype 2 may have a more distinct immune profile.

Immunotherapy is an emerging anti-tumor treatment that allows the immune system to be manipulated to recognize and attack cancer cells [87]. ICIs, as part of immunotherapy, achieve the process of anti-tumor immunity primarily by disrupting negative immunomodulatory checkpoints and releasing pre-existing anti-tumor immune responses [88]. A higher TMB could predict the efficacy of ICIs, primarily because high TMB released more neoantigens and increased the chance of being recognized by T cells [89]. However, a recent ACC-based study on the relationship between TMB and ICIs pointed out that there may be no difference in the tumor immune microenvironment between high- and low-TMB groups, and that TMB did not affect the activity of immune-related pathways and predict the response to ICIs [90]. Combined with our findings, although there was a significant TMB difference between subtypes 1 and 2, its role in the prediction of ICIs was not well represented, indicating that TME-related subtypes may not predict ICIs response through TMB. In addition to TMB, ICIs and MHC molecules can also be used as biomarkers of ICIs to predict patients’ response to ICIs. MHC molecules primarily include MHC-I and MHC-II; killing tumor cells by CD8^+^ T cells required the effective presentation of tumor antigens by human leukocyte antigen class I (HLA-I) molecules [91]. Expression of MHC-II molecules was one of the key factors in the response to ICIs and may be associated with the recognition of tumor-specific antigens by helper CD4^+^ T cells during ICIs and the promotion of tumor inhibition [92]. More MHC molecules were observed in subtype 2, suggesting that subtype 2 may be associated with the promotion of immune response.

Although subtype 2 expressed more ICIs relative to subtype 1, the applicability of ICIs may not be completely uniform due to tumor differences. For example, overexpression of PD-L1 molecules could promote immune escape by suppressing CTLs function, while PD-L1 expression was also accompanied with IFN-γ production by tumor infiltrating T cells [93]. Therefore, overexpression of PD-L1 levels may not be a robust predictor of ICI response [94]. Likewise, PD-1 expression may not be a good diagnostic indicator in all patients treated with ICIs [95]. Therefore, a single ICs to determine whether TME-related subtypes had some application value in immunotherapy remains limited. To further confirm whether TME-related subtypes were predictive of response to immunotherapy, we used more widely used immunotherapy prediction algorithms: TIDE and IPS. Although there were several algorithms that could be used to predict response to immunotherapy, these two algorithms were somewhat more comprehensive in their application when compared with others. First, the IPS algorithm considered both the expression of immune-related molecules and the marker genes of immune cells. The TIDE algorithm was not limited to the expression of a few key genes, but systematically considered the prognostic impact of the interaction of all the genes and CTLs on patients. From our prediction results, patients with subtype 2 had lower TIDE scores and higher IPS scores, indicating that subtype 2 was more advantageous for immunotherapy.

Effective risk stratification and survival prediction were important for the precise management of ACC. Although there were many clinical indicators that could identify risk and effectively predict survival in patients with ACC, there was still a need to develop other biomarkers for the purpose of rationalizing appropriate interventions for different risk groups. We classified ACC into two different subtypes based on the expression of TME-related genes and initially determined that these two subtypes were distinct in terms of immune response. To further identify candidate TME-related genes, we used the TME-related subtype as the main feature and identified modules significantly associated with TME-related subtypes using the WGCNA method. Ultimately, we obtained key modules containing 271 genes, and we found that they were enriched in functions, primarily including response to interferon gamma, MHC protein complexes, and immune receptor activity. The signaling pathways primarily included complement and coagulation cascade and Allograft rejection. These results were consistent with the differential pathways associated with subtype 2, indicating that these genes in the key module were not only correlated in expression with TME-related subtypes but also had some similarity in biological functions, further suggesting that these genes could be valuable as TME-related subtype genes. Therefore, we intersected these genes and TME-related genes as candidate TME-related genes. Based on these candidate TME-related genes, we established a TME-related signature for ACC using univariate Cox and LASSO regression analyses.

Some of the genes in the model may influence tumor progression by cell death. For example, GBP2 activated caspase-4, that triggered cell death and inhibited tumor cell proliferation [96,97], and SLC40A1, that acted as a negative regulator of ferroptosis. However, ACC showed a high sensitivity to induced ferroptosis [98,99], so SLC40A1 may influence ACC progression by regulating ferroptosis. In contrast, TFEB regulatedcellular autophagy by controlling gene expression; autophagy has been shown to promote or inhibit ACC [100,101]. In addition to the genes involved in the programmed cell death process, other genes also influenced the prognosis of ACC patients through corresponding pathways. CX3CR1 has some ability to distinguish between survival and death states in ACC, and its low expression was associated with a higher OS [102]. CYTL1 has been reported to inhibit STAT3 phosphorylation, and a decrease in STAT3 phosphorylation expression was accompanied by an increase in anti-angiogenic effects, thereby inhibiting tumor progression [103,104,105]. A pan-cancer study found that RGS1 was associated with prognosis in multiple tumors, with its high expression leading to the development of a poorer prognosis in ACC, as well as a facilitative effect on T cell exhaustion [106]. Through internal and external validation of the TME-related signature, we found that this signature has an excellent risk identification and survival prediction ability, and remained prognostic even when integrated with other clinical indicators, indicating that our established risk score had an independent predictive value in the existing datasets.

In the present study, we conducted a more in-depth analysis, primarily focusing on the fact that we not only discussed the impact of TME-related characteristics on ACC but also constructed two TME-related subtypes of ACC and revealed their potential in immunotherapy from different perspectives. Thus, the final constructed TME-related signature differed from previous similar signatures because we fully considered the biological significance brought by TME-related subtypes. However, there were still several shortcomings in this study. First, although we estimated the TME-related scores of ACC using the xCell algorithm, further estimation of TME by means of single-cell sequencing technology is still required. Second, although we initially determined that TME-related subtypes could be used as a predictive indicator for immunotherapy in this study, further validation in combination with a real immunotherapy cohort is needed in the future. Finally, more samples with high heterogeneity need to be included for further training and validation in future studies though the TME-related signature we constructed had a high predictive potential in the prognosis of ACC.

## 5. Conclusions

In this study, we quantified the TME of ACC using a bioinformatics approach. This helped to explain the overall effect of TME on patient prognosis and revealed the potential application of TME-related subtypes in immunotherapy. The established TME-related signature was beneficial for risk intervention and prognosis prediction, to a certain extent. We believe that our study can provide valuable strategies for future clinical treatment of ACC.

## Figures and Tables

**Figure 1 cells-12-00755-f001:**
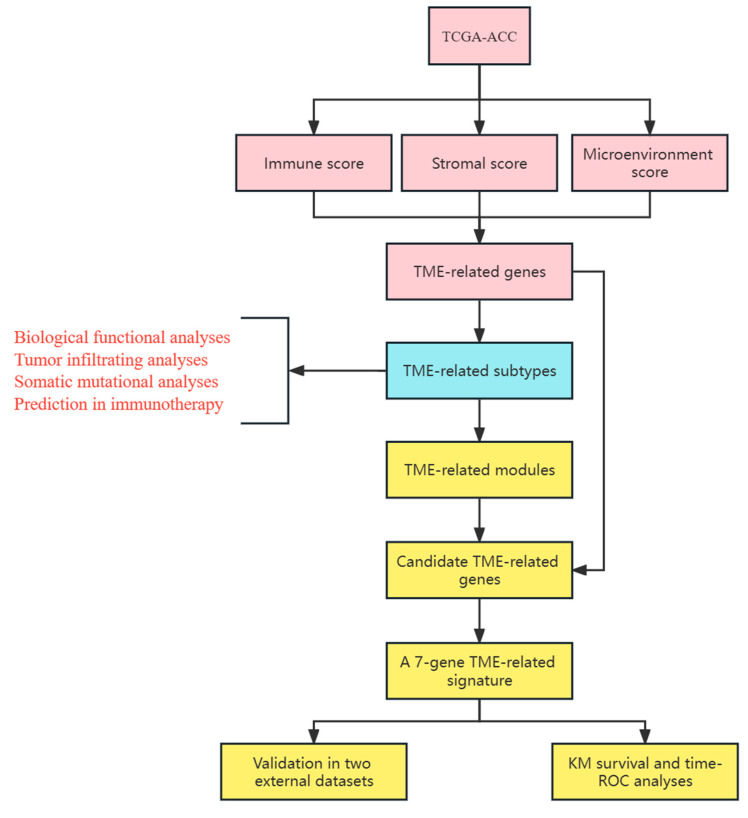
The flow chart of the study.

**Figure 2 cells-12-00755-f002:**
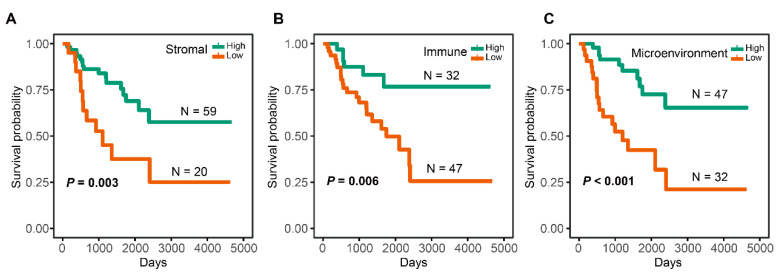
The relationship between TME-related scores and OS. (**A**) KM curves for patients with high- and low-stromal scores. (**B**) KM curves for patients with high- and low-immune scores. (**C**) KM curves for patients with high- and low-microenvironment scores.

**Figure 3 cells-12-00755-f003:**
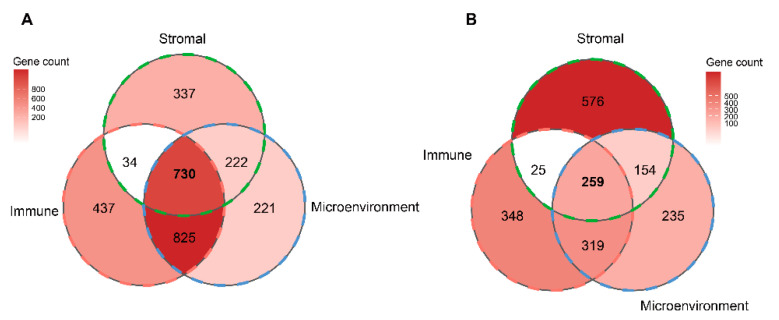
The determination of TME-related genes. (**A**) Venn diagram showing the number of DEGs that were up-regulated in the immune, stromal and, microenvironment groups. (**B**) Venn diagram showing the number of DEGs that were down-regulated in the immune, stromal, and microenvironment groups.

**Figure 4 cells-12-00755-f004:**
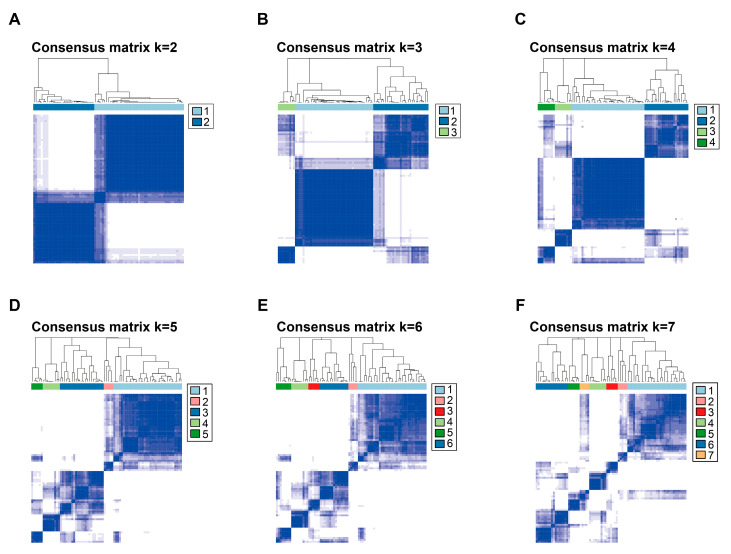
Consensus clustering matrix of TCGA-ACC samples. (**A**) Consensus clustering matrix of TCGA-ACC samples for k = 2. (**B**) Consensus clustering matrix of TCGA-ACC samples for k = 3. (**C**) Consensus clustering matrix of TCGA-ACC samples for k = 4. (**D**) Consensus clustering matrix of TCGA-ACC samples for k = 5. (**E**) Consensus clustering matrix of TCGA-ACC samples for k = 6. (**F**) Consensus clustering matrix of TCGA-ACC samples for k = 7.

**Figure 5 cells-12-00755-f005:**
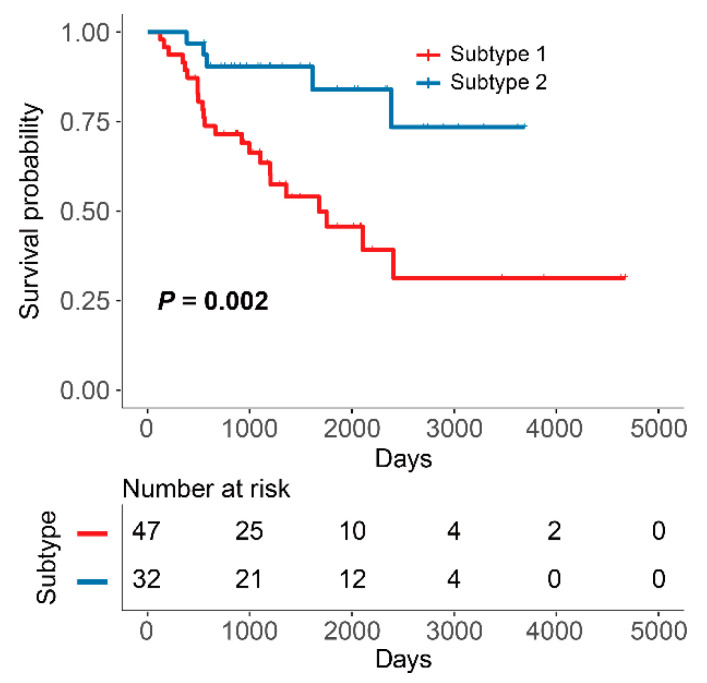
KM curves for OS of patients with subtypes 1 and 2.

**Figure 6 cells-12-00755-f006:**
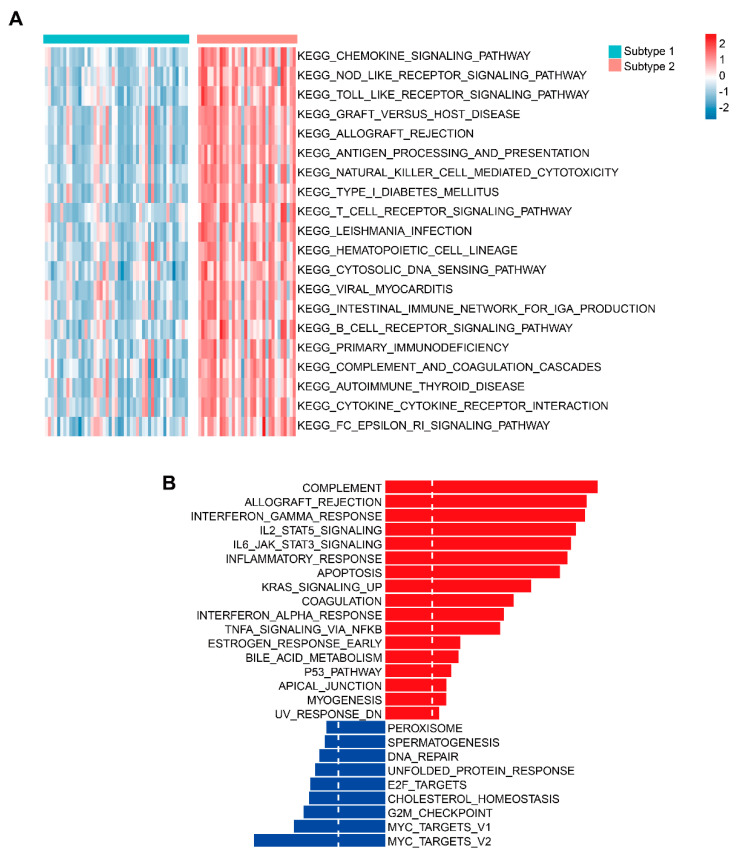
GSVA analysis of KEGG and Hallmark pathways for TME-related subtypes. (**A**) Heatmap showing the top 20 most significant KEGG pathways between subtypes 1 and 2. (**B**) Distribution of significant Hallmark pathways between subtypes 1 and 2.The red item represented the most relevant pathways to subtype 2. The blue item represented the most relevant pathways to subtype 1.

**Figure 7 cells-12-00755-f007:**
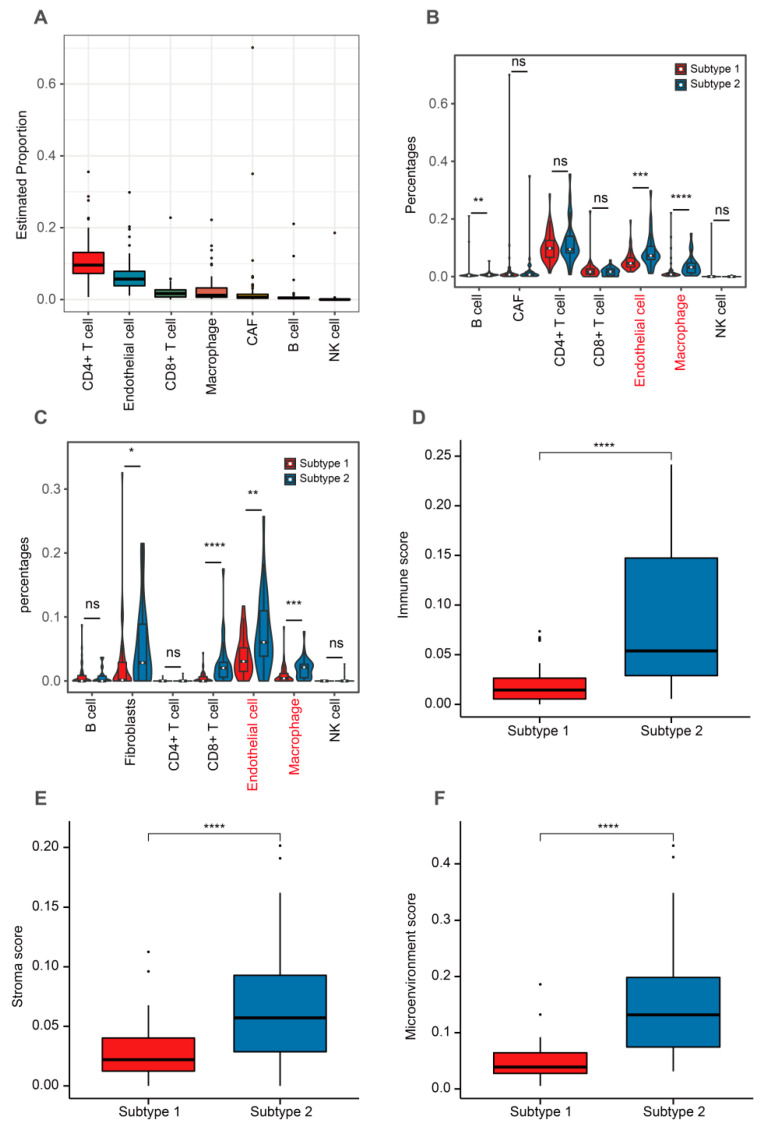
Comparison of tumor infiltrating features between two TME-related subtypes. (**A**) The estimated proportions of seven cells according to the EPIC algorithm. (**B**) Comparison of the proportions of these seven cells calculated through the EPIC algorithm between two subtypes. (**C**) Comparison of the proportions of these seven cells calculated through the xCell algorithm between two subtypes. (**D**–**F**) Comparison of TME-related scores between subtypes 1 and 2. Data in (**B**–**F**) were analyzed by Wilcoxon test; ns: No significance, * *p* < 0.05, ** *p* < 0.01, *** *p* < 0.001 and **** *p* < 0.0001.

**Figure 8 cells-12-00755-f008:**
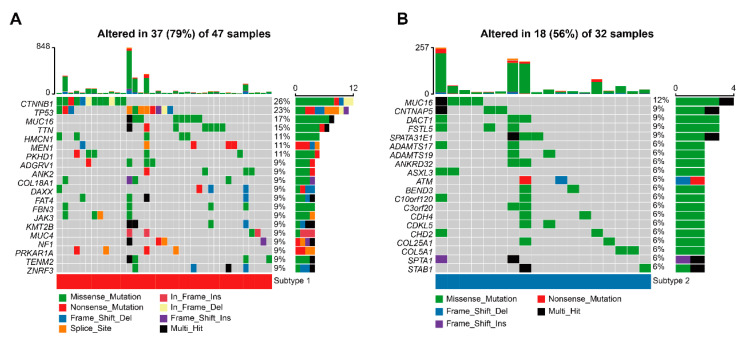
Somatic variants and TME-related subtypes. (**A**) Waterfall plot visualizing the top 20 highly mutated genes of subtype 1. (**B**) Waterfall plot visualizing the top 20 highly mutated genes of subtype 2.

**Figure 9 cells-12-00755-f009:**
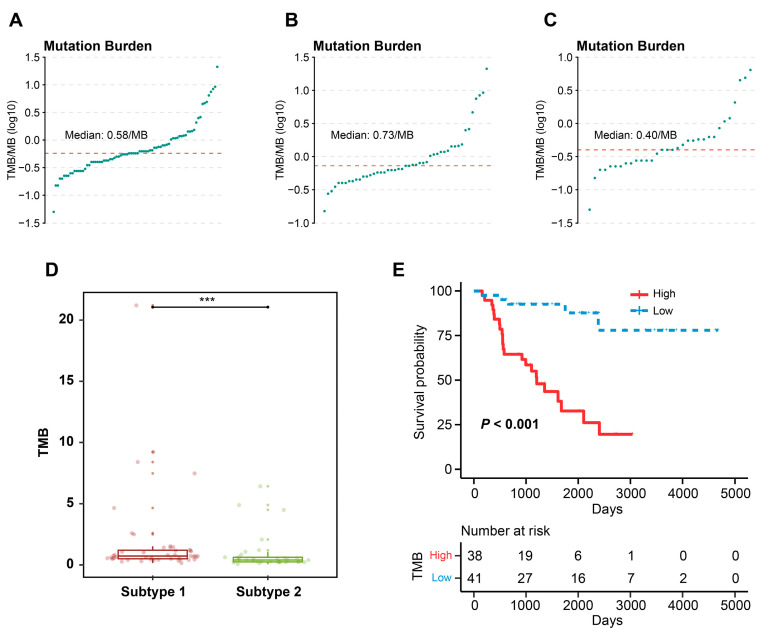
TMB profile and TME-related subtypes. (**A**) The distribution of TMB of TCGA-ACC patients. (**B**) The distribution of TMB of subtype 1. (**C**) The distribution of TMB of subtype 2. (**D**) Comparison of the differences in TMB between two subtypes. (**E**) KM curves of OS for high- and low-TMB groups. Data in (**D**) was analyzed by Wilcoxon test; *** *p* < 0.001.

**Figure 10 cells-12-00755-f010:**
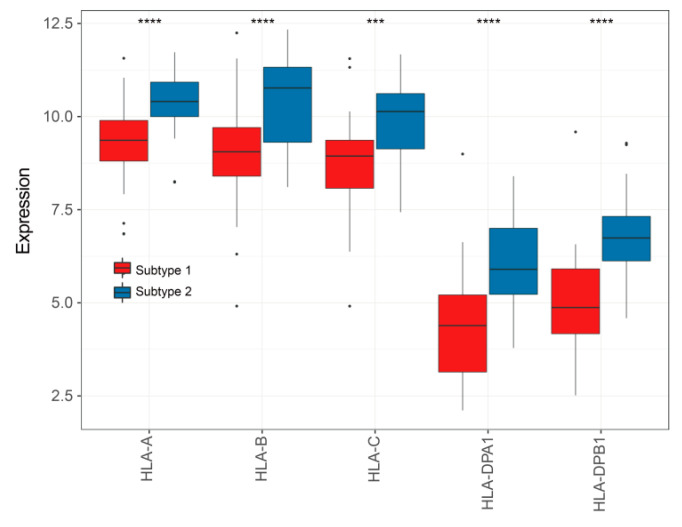
Comparisons of five MHC molecules betweensubtypes1 and 2. Data was analyzed using the Wilcoxon test; *** *p* < 0.001, **** *p* < 0.0001.

**Figure 11 cells-12-00755-f011:**
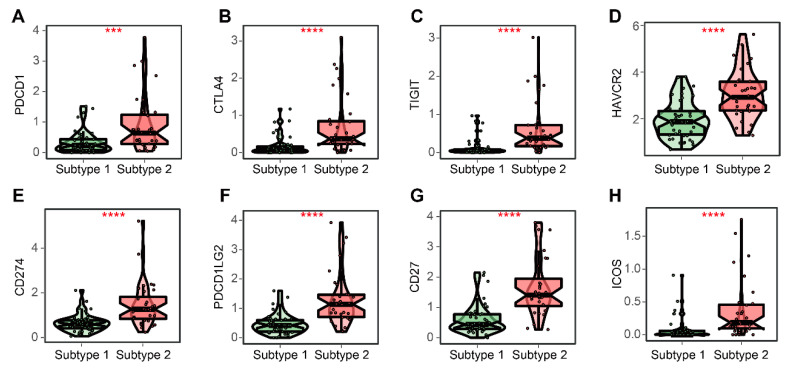
Comparisons of eight ICs between subtypes 1 and 2. (**A**–**H**) Expressions of PDCD1, CTLA4, TIGIT, HAVCR2, CD274, PDCD1LG2, CD27, and ICOS between subtypes 1 and 2. Data in (**A**–**H**) were analyzed using the Wilcoxon test; *** *p* < 0.001, **** *p* < 0.0001.

**Figure 12 cells-12-00755-f012:**
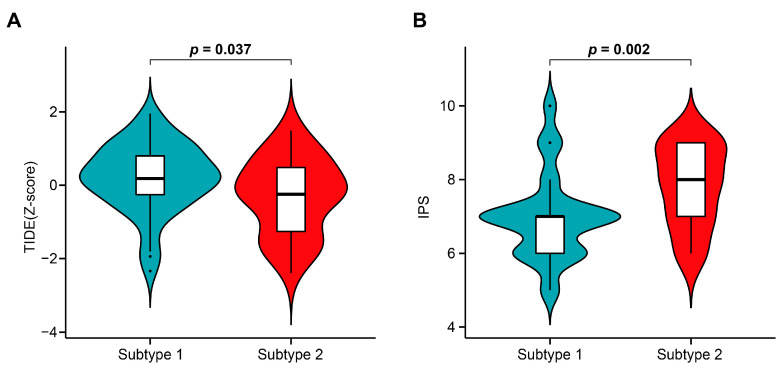
Association of TME-related subtypes with immunotherapy. (**A**) Comparisons of normalized TIDE scores between subtypes 1 and 2. (**B**) Comparison of IPS between subtypes 1 and 2.

**Figure 13 cells-12-00755-f013:**
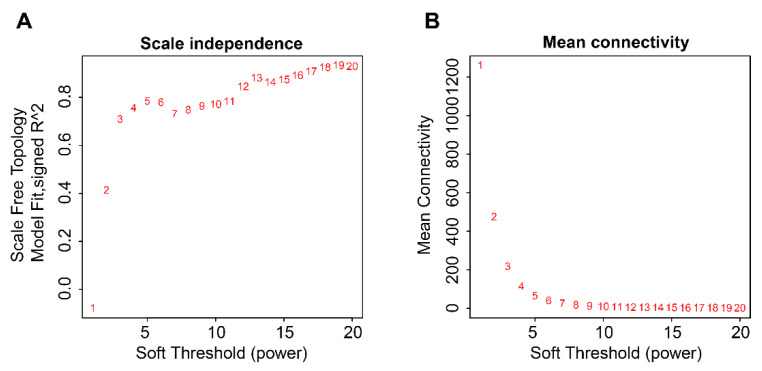
The process of determining an optimal soft threshold. (**A**) Change of the scale-free topological fit index R^2^ for different soft threshold. (**B**) Change of the mean connectivity for different soft threshold.

**Figure 14 cells-12-00755-f014:**
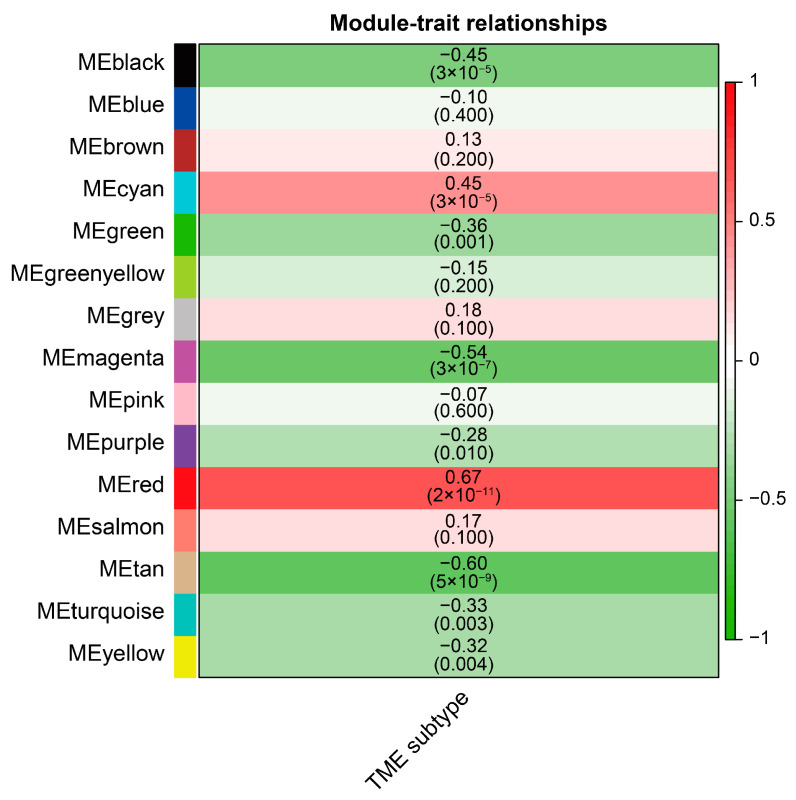
Heatmap showing the correlation between each module eigengenes and TME-related subtypes.

**Figure 15 cells-12-00755-f015:**
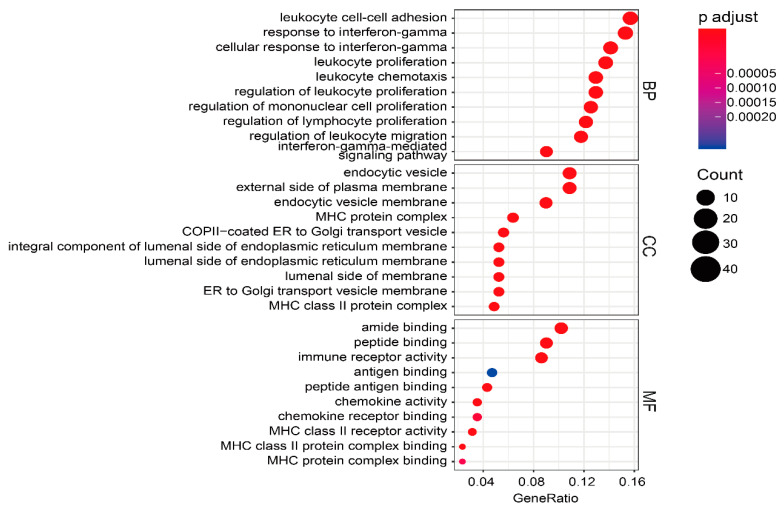
GO enrichment analysis of 271 TME-related subtype genes, with the 10 most significant functions in each panel.

**Figure 16 cells-12-00755-f016:**
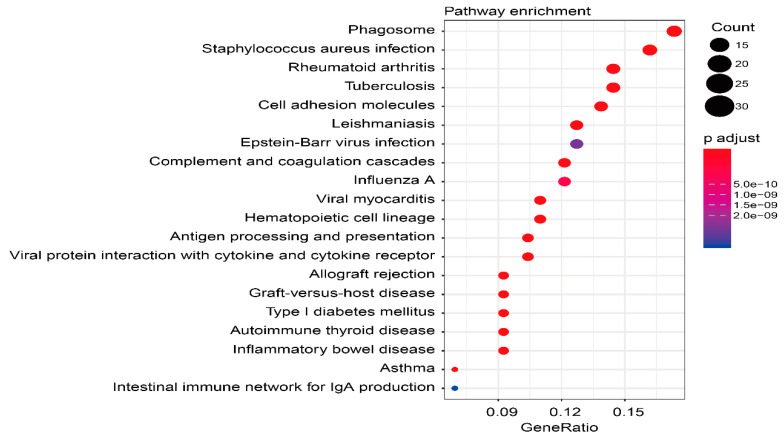
KEGG pathway analysis of 271 TME-related subtype genes, with the 10 most significant pathways.

**Figure 17 cells-12-00755-f017:**
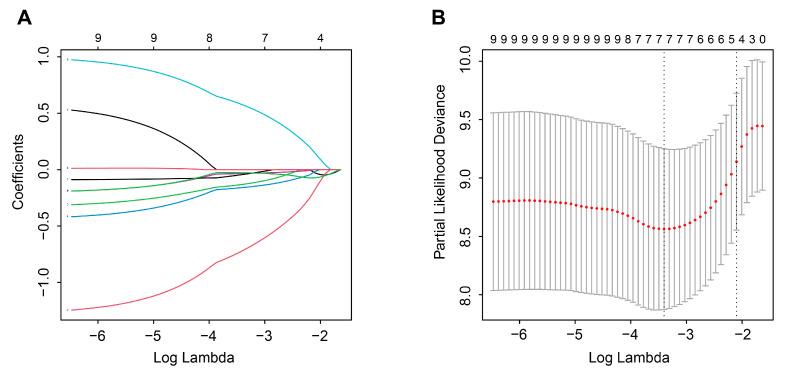
Screening prognostic genes by LASSO regression model. (**A**) Regression coefficient maps of 9 genes under different λ. (**B**) 10-fold cross-validation to determine λ.

**Figure 18 cells-12-00755-f018:**
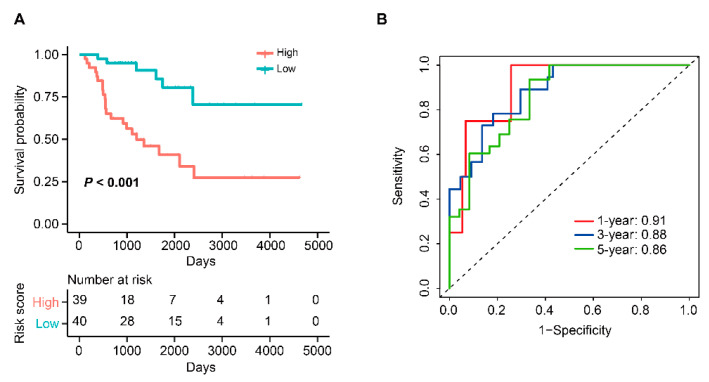
Identification of a novel TME-related signature in TCGA-ACC dataset. (**A**) KM curves of OS for patients in the high- and low-risk groups. (**B**)Time-ROC curve to predict the OS at year 1, 3, and 5.

**Figure 19 cells-12-00755-f019:**
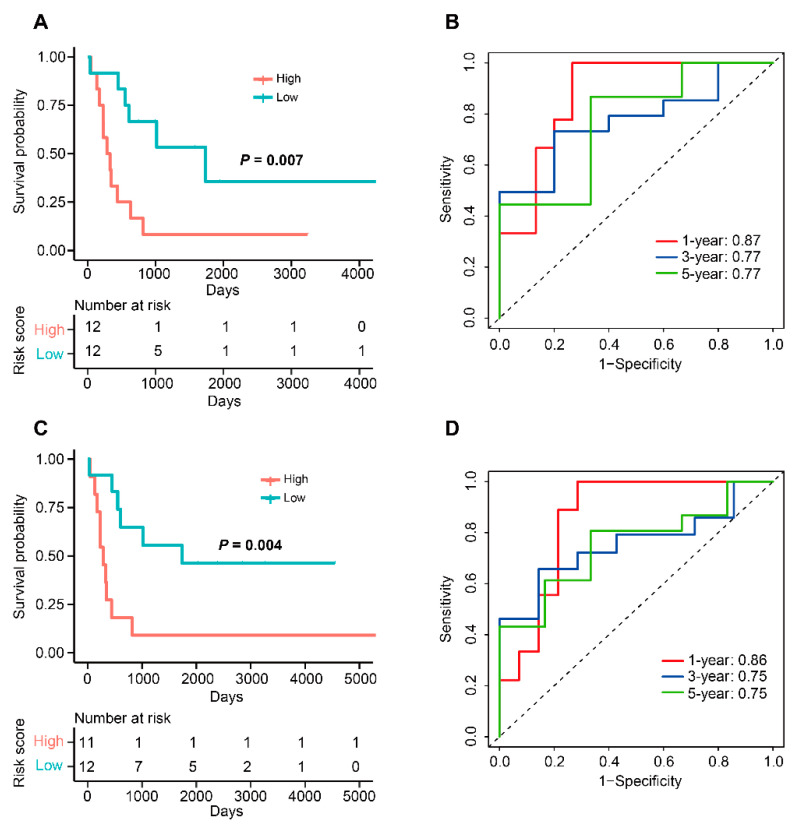
Validation of this TME-related signature in two external datasets. (**A**) KM curves of OS for patients in the high- and low-risk groups in GSE10927. (**B**)Time-ROC curve to predict the OS at year 1, 3, and 5 in GSE10927. (**C**) KM curves of OS for patients in the high- and low-risk groups in GSE33371. (**D**)Time-ROC curve to predict the OS at year 1, 3, and 5 in GSE33371.

**Table 1 cells-12-00755-t001:** The results of univariate and multivariate Cox analyses in TCGA-ACC dataset.

Variable	Univariate Analysis		Multivariate Analysis	
	HR	*p* Value	HR	*p* Value
Risk score	4.10 (2.36–7.13)	<0.001	3.06 (1.73–5.40)	<0.001
Age	1.01 (0.99–1.04)	0.379		
Gender (Male vs. Female)	1.00 (0.47–2.14)	0.999		
M stage (M1 vs. M0)	6.15 (2.71–13.96)	<0.001	1.12 (0.42–3.0)	0.813
N stage (N1 vs. N0)	2.04 (0.77–5.40)	0.152		
T stage (T3-T4 vs. T1-T2)	10.29 (3.98–26.61)	<0.001	5.79 (0.71–47.40)	0.102
Stage (III-IV vs I-II)	6.48(2.71–15.50)	<0.001	0.75 (0.09–6.20)	0.790

**Table 2 cells-12-00755-t002:** The results of univariate Cox analyses in two external datasets.

Variable	GSE33371		GSE10927	
	HR	*p* Value	HR	*p* Value
Risk score	1.91 (1.16–3.13)	0.011	1.95 (1.19–3.20)	0.008
Age	1.01 (0.96–1.06)	0.775	1.01 (0.96–1.06)	0.639
Gender (Male vs. Female)	1.36 (0.47–3.95)	0.574	1.47 (0.51–4.24)	0.475
Weiss score (Low vs. High)	0.32 (0.09–1.15)	0.082	0.35 (0.10–1.24)	0.104
Stage (III–IV vs. I–II)	2.56 (0.92–7.10)	0.071	2.56 (0.96–6.83)	0.060

## Data Availability

The results shown here are based on data generated by TCGA (https://xenabrowser.net/, accessed on 1 November 2022), GEO (http://www.ncbi.nlm.nih.gov/geo/, accessed on 3 November 2022), Ensembl (http://Asia.ensembl.org/, accessed on 2 November 2022) and MSigDB (https://www.gsea-msigdb.org/, accessed on 7 November 2022) databases.

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
