# Peer review of "The Characteristics of Tumor Microenvironment Predict Survival and Response to Immunotherapy in Adrenocortical Carcinomas"

_cells, 2023, doi:10.3390/cells12050755_

Round 1

Reviewer 1 Report

1)      In figure 3, authors should describe or present in the supplementary table which genes are up-regulated and which genes are down-regulated. Is there any possibility that a unique gene is simultaneously down-regulated and up-regulated at different times?

2)      Figure 4 is not good at all. It should be re-drawn with a higher resolution. Why the matrix is hidden the behind the images? It must be labeled as A, B… and described in the figure’s legend.

3)      All of the figures should be self-informative. On the other hand, readers should be able to understand the content of the figure from the legend. Unfortunately, most of the figures are not like this, such as figs 4, 5, 10, 14 …

4)      Why in figure 12, these two P-values ( P= 0.037 & P=0.002) have been selected?

5)      The authors must describe why some error bars are very large like figs 10 and 7E&F. 

Reviewer 2 Report

In this paper, Lai and collaborators analyzed by bioinformatic tools existing gene expression databases of adrenocortical carcinma (ACC) to investigate the composition of the tumor microeenvironment (TME) and to assay its prognostic significance. A TME-derived signature was identified which had prognostic value and may help to identify patients with ACC who can respond to immunotherapy.

This is an interesting study which opens potential new perspectives in the therapy of ACC. However, several points should be considered:

1) - page 2, line 61 and following: this is the Znrf3 KO mouse model, which is not relevant to human ACC. Delete this reference, which is present twice in the reference list (#27 and 80).

 2) - page 3, line 106: do the authors mean "cells" or "cell types" here?

 3) - page 6, lines 231-232: define CDF and PAC

 4) - page 8, line 263: again, do the authors mean "cells" or "cell types" here?

5) - page 8-9, paragraph 3.6 and Fig. 7: are subtypes 1 and 2 here the same as in Fig. 5? Probably colors are inverted. They should be made the same.

6) - page 10, paragraph 3.7: doesn’t it seem paradoxical that tumors with high TMB have a lower immune score? Please comment.

7) - Fig. 10: again, make colors uniform: red subtype 1, blue subtype 2

8) - page 12, paragraph 3.8: the 271 red module genes should be listed in a Supplementary Table.

Author Response

Dear reviewer,

Thank you for reviewing our manuscript titled “The characteristics of tumor microenvironment predict survival and response to immunotherapy in adrenocortical carcinomas”. We have carefully studied the comments and suggestions, and then revised the manuscript accordingly. The changed and added texts in the manuscript are shown in yellow. We hope that the revision could be acceptable, and that our responses adequately address the comments. Should you have any questions, please contact us without hesitation.

Q1 - page 2, line 61 and following: this is the Znrf3 KO mouse model, which is not relevant to human ACC. Delete this reference, which is present twice in the reference list (#27 and 80).

Response: Thanks for your suggestion. We have deleted this reference and added a new reference to support the fact that patients with subtype 2 had higher infiltrating levels of macrophages than patients with subtype 1 in the revised manuscript (Line 501-504, Page 19).

Q2 - page 3, line 106: do the authors mean "cells" or "cell types" here?

Response: Thanks for your comment. We referred to the cell types here.

Q3 - page 6, lines 231-232: define CDF and PAC

Response: Thanks for your suggestion. We have defined the CDF and PAC in the revised manuscript (Line 127, Page 4; Line 235-236, Page 6).

Q4 - page 8, line 263: again, do the authors mean "cells" or "cell types" here?

Response: Thanks for your comment. We referred to the cells here.

Q5 - page 8-9, paragraph 3.6 and Fig. 7: are subtypes 1 and 2 here the same as in Fig. 5? Probably colors are inverted. They should be made the same.

Response: Thanks for your suggestion. Subtypes 1 and 2 in Figure 7 were same as them in Figure 5. We have unified their colors in revised manuscript (Line 251-252, Page 7; Line 281-282, Page 9).

Q6 - page 10, paragraph 3.7: doesn’t it seem paradoxical that tumors with high TMB have a lower immune score? Please comment.

Response: Thanks for your comment. Indeed, although high tumor mutational load predicts the efficacy of immune checkpoint inhibitors (ICIs), this is not true for all tumors. For adrenocortical carcinoma (ACC), there are no experimental studies to confirm whether high TMB is equally predictive of response to ICIs in ACC. A recent ACC-based study on the relationship between TMB and ICIs pointed out that there may be no difference in tumor immune microenvironment between high- and low-TMB groups, and TMB did not affect the activity of immune-related pathways and predict the response to ICIs [1]. Wang et al. found that ACC with low TMB exhibited higher levels of tumor-infiltrating lymphocytes, HLA genes expression, cytokine-related genes expression, and pro-inflammatory genes expression relative to the high-TMB group, suggesting that high TMB may be associated with immunosuppression [2].

Q7 - Fig. 10: again, make colors uniform: red subtype 1, blue subtype 2

Response: Thanks for your suggestion. We have unified their colors in revised manuscript (Line 329, Page 12).

Q8 - page 12, paragraph 3.8: the 271 red module genes should be listed in a Supplementary Table.

Response: Thanks for your suggestion. 271 red module genes have been listed in the Supplementary Table 6 (Line 633, Page 21).

References:

  1. Xu, F.; Guan, Y.; Zhang, P.; Xue, L.; Ma, Y.; Gao, M.; Chong, T.; Ren, B. C. Tumor mutational burden presents limiting effects on predicting the efficacy of immune checkpoint inhibitors and prognostic assessment in adrenocortical carcinoma.  Endocr. Disord. 2022, 22, 130. doi:10.1186/s12902-022-01017-3
  2. Wang, X.; Li, M. Correlate tumor mutation burden with immune signatures in human cancers.  Immunol. 2019, 20, 4. doi:10.1186/s12865-018-0285-5

Round 2

Reviewer 1 Report

The authors answered my comments. 

Author Response

Thank you for your review and response!